 eLIFE

# A serine sensor for multicellularity in a bacterium

**Arvind R Subramaniam[1,2], Aaron DeLoughery[1], Niels Bradshaw[1], Yun Chen[1], Erin O'Shea[1,2,3,4], Richard Losick[1]\*, Yunrong Chai[1,5]\***

[1]Department of Molecular and Cellular Biology, Harvard University, Cambridge, United States; [2]Faculty of Arts and Sciences Center for Systems Biology, Harvard University, Cambridge, United States; [3]Department of Chemistry and Chemical Biology, Harvard University, Cambridge, United States; [4]Howard Hughes Medical Institute, Harvard University, Cambridge, United States; [5]Department of Biology, Northeastern University, Boston, United States

**Abstract** We report the discovery of a simple environmental sensing mechanism for biofilm formation in the bacterium *Bacillus subtilis* that operates without the involvement of a dedicated RNA or protein. Certain serine codons, the four TCN codons, in the gene for the biofilm repressor SinR caused a lowering of SinR levels under biofilm-inducing conditions. Synonymous substitutions of these TCN codons with AGC or AGT impaired biofilm formation and gene expression. Conversely, switching AGC or AGT to TCN codons upregulated biofilm formation. Genome-wide ribosome profiling showed that ribosome density was higher at UCN codons than at AGC or AGU during biofilm formation. Serine starvation recapitulated the effect of biofilm-inducing conditions on ribosome occupancy and SinR production. As serine is one of the first amino acids to be exhausted at the end of exponential phase growth, reduced translation speed at serine codons may be exploited by other microbes in adapting to stationary phase.

**\*For correspondence:**
losick@mcb.harvard.edu (RL);
y.chai@neu.edu (YC)

**Reviewing editor**: Gisela Storz, National Institute of Child Health and Human Development, United States

## Introduction

Bacteria constantly monitor their environment and internal physiological state so that they can adapt to changing conditions. A wide variety of sensing mechanisms are deployed for this purpose, including dedicated protein sensors, such as histidine kinases, which mediate changes in gene expression by controlling the phosphorylation of cognate response regulators in response to environmental cues (*West and Stock, 2001*). Bacteria also sense changes in their environment and physiology by means of dedicated RNAs, such as the highly structured, leader RNA for the tryptophan operon, which controls the transcription of downstream genes in the operon by a mechanism involving ribosome stalling at tryptophan codons (*Henkin and Yanofsky, 2002*). Here we report the discovery of an unusually simple mechanism of environmental sensing involved in the process of biofilm formation by the bacterium *B. subtilis* that does not require a dedicated RNA or protein.

Biofilm formation involves a switch from planktonic growth as individual cells to the formation of complex, multicellular communities in response to environmental cues (*Kolter and Greenberg, 2006*). In *B. subtilis*, these communities are embedded in a self-produced matrix consisting of polysaccharide and an amyloid-like protein, which are specified by the *epsA-O* and the *tapA-sipW-tasA* operons, respectively (*Branda et al., 2001*; *Kearns et al., 2005*). The transition to multicellularity is governed in part by four histidine kinases (KinA, KinB, KinC and KinD) that control the phosphorylation of the response regulator, Spo0A, a master regulator of post-exponential phase gene expression (*Figure 1A*) (*Jiang et al., 2000*; *Vlamakis et al., 2013*). Recent studies suggest that KinA and KinB respond to impaired respiration (*Kolodkin-Gal et al., 2013*), whereas KinC responds to membrane perturbations

**eLife digest** Bacteria use several different mechanisms to recognize and respond to changes in their environment. Protein sensors, for example, relay signals from the cell surface to target molecules within the cell. Additionally, RNA sensors can respond to internal levels of chemicals by regulating the expression of genes. Now, Subramaniam et al. have discovered a sensing mechanism in bacteria that does not rely on either protein sensors or RNA sensors.

Bacteria normally live as free swimming cells in water. But sometimes, in response to certain environmental conditions, they form a multicellular community called a biofilm. In this biofilm, bacteria encase themselves with layers of carbohydrates and proteins, which then protect the bacteria from adverse chemicals such as antibiotics.

A protein known as SinR plays a key role during biofilm formation in the model bacterium *Bacillus subtilis.* SinR normally prevents the formation of the carbohydrates and proteins that make up the biofilm. Upon the decision to form a biofilm, *B. subtilis* counters the effect of SinR by producing an anti-SinR protein called SinI.

Now Subramaniam et al. have found that as well as producing the SinI protein, *B. subtilis* use an additional mechanism to promote biofilm formation. This mechanism relies on codons, the elements within genes that correspond to specific amino acids. Six different codons correspond to the amino acid serine, and the gene for the SinR protein contains above average numbers of four of them. These four codons are highly sensitive to serine levels, and they decrease the levels of the SinR protein when there is less serine in the environment, as happens to be the case in biofilms. And since SinR prevents the production of the carbohydrates and proteins that make up the biofilms, a decrease in the levels of SinR leads to an increase in the production of biofilms.

Since the serine codons at the heart of the sensing mechanism discovered by Subramaniam et al. are present in all forms of life, from viruses to humans, it is possible that similar sensing mechanisms might be found in contexts other than bacterial biofilms, such as in viral infection and cancer.

and KinD to unknown chemical signals (*López et al., 2009*; *Shemesh et al., 2010*; *Chen et al., 2012*; *Beauregard et al., 2013*). Once phosphorylated, Spo0A turns on *sinI*, a gene encoding a small protein antagonist of the biofilm-specific regulatory protein SinR (*Molle et al., 2003*; *Kearns et al., 2005*). SinR, which is produced constitutively, is a repressor of the matrix operons, *epsA-O* and the *tapA-sipW-tasA*, as well as other biofilm-related genes (*Kearns et al., 2005*; *Chu et al., 2006*; *Chai et al., 2009*). SinR is also a repressor of the gene for SlrR (*Chu et al., 2008*), which together with SinR sets up a self-reinforcing, double-negative feedback loop for matrix gene expression (*Figure 1A*) (*Chai et al., 2010*; *Norman et al., 2013*). A special feature of SinR of relevance to this investigation is that the expression of matrix genes is hypersensitive to small perturbations in the level of the protein (*Chai et al., 2011*). This hypersensitivity is attributed to molecular titration of SinR by SinI and cooperativity among SinR molecules bound to tandem target sequences at regulatory sites for the matrix operons (*Chai et al., 2011*; *Chai et al., 2008*).

In the current study, we present evidence for the existence of a novel cellular sensing mechanism controlling biofilm formation. Rather than relying on regulation by a dedicated RNA or protein, the translation speed of ribosomes decreases at certain serine codons, resulting in lower SinR levels, which as a consequence, contributes to derepression of SinR-controlled genes. We propose that specific serine codons in the *sinR* mRNA act as a simple sensor for monitoring, and triggering a response to, serine depletion under biofilm-inducing conditions.

## Results

### Switching synonymous serine codons in *sinR* affects biofilm formation

In a genetic screen to identify suppressor mutations that rescued the biofilm-defective phenotype of a *B. subtilis* mutant ('Materials and methods'), we unexpectedly recovered a variant that contained a 'silent' mutation that resulted in a switch from one serine codon to a synonymous codon in *sinR*. This observation prompted us to ask whether switching other serine codons might also influence biofilm formation. Serine is specified by six codons: AGC, AGT, TCA, TCC, TCG and TCT (where T is U in

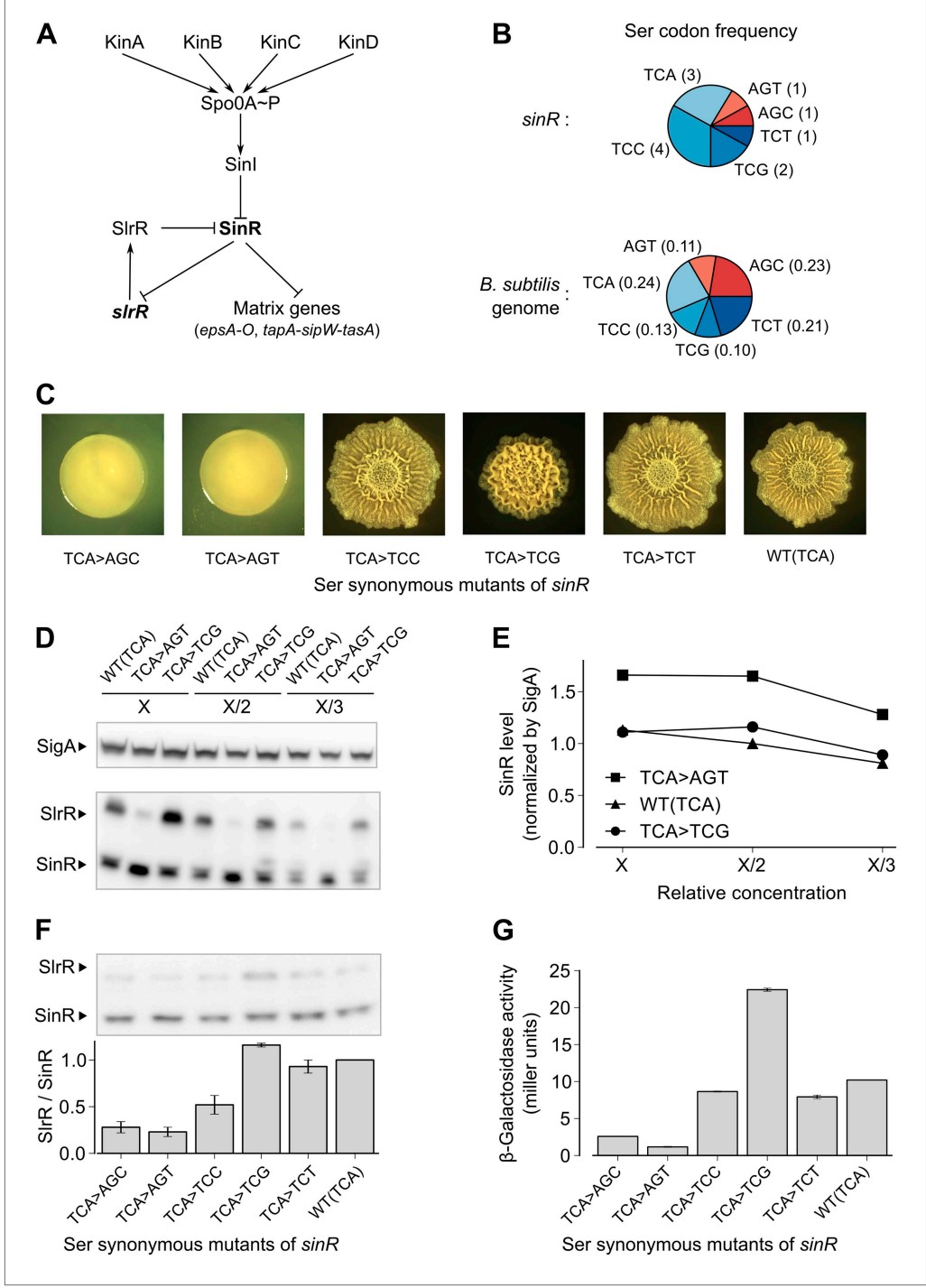

**Figure 1**. Switching synonymous serine codons in *sinR* affects biofilm formation. (**A**) Regulatory circuit controlling biofilm formation in *B. subtilis.* (**B**) Top: Serine codon usage in the *sinR* coding sequence. Number within parenthesis indicates the frequency of the corresponding codon in *sinR*. Bottom: Average serine codon usage across 4153 protein-coding sequences in the *B. subtilis* genome. Number within parenthesis indicates the relative frequency of each codon in the genome. (**C**) Colony morphology for the wild-type strain and the indicated *sinR* synonymous variants grown on solid biofilm-inducing medium. Three TCA codons in the wild-type sequence of *sinR* were switched to each of the other five serine codons. The wild-type (WT) *sinR* sequence was replaced by the *sinR* synonymous mutant at the native *sinR* locus of the strain 3610. (**D**) SinR protein level during entry into biofilm formation ($OD_{600} = 2$) measured using an anti-SinR antibody that also cross-reacts with SlrR, a protein that is

*Figure 1. Continued on next page*

*Figure 1. Continued*

85% identical to SinR. Western blot against the RNA polymerase subunit SigA was used as the loading control. Whole cell lysates were loaded at different dilutions (indicated as X, X/2, and X/3). (**E**) Densitometry of SinR bands in (**D**) after normalization by SigA. (**F**) Top panel: Western blot against SinR and SlrR using anti-SinR antibody. Bottom panel: Densitometry ratio of the SlrR and SinR bands in the top panel. Error bars represent standard error over three replicate Western blots. The SlrR/SinR ratio for each blot was normalized such that the wild-type strain had a ratio of 1. (**G**) Matrix gene expression monitored using a $P_{epsA}$–*lacZ* transcriptional reporter inserted at the chromosomal *amyE* locus. *β*-galactosidase activity was measured at $OD_{600}$ = 2 in liquid biofilm-inducing medium. Error bars represent standard error of three measurements.

The following figure supplements are available for figure 1:

**Figure supplement 1**. *sinR* coding sequence.

**Figure supplement 2**. Effect of TCC and AGC/AGT synonymous substitutions in the *sinR* gene on colony morphology and biofilm reporter activity.

the mRNA). We noticed that the *sinR* coding sequence has a slightly higher frequency of the four TCN serine codons as compared to the average frequency of TCN codons in the *B. subtilis* genome (*Figure 1B*, p=0.22, N = 12). To test whether this bias towards TCN codons has an effect on biofilm formation, we systematically replaced the three TCA codons in *sinR* (*Figure 1—figure supplement 1*) with each of the other five serine codons. Replacing the TCA codons with AGT or AGC resulted in flat, featureless colonies on solid biofilm-inducing medium, indicating severely impaired biofilm formation (*Figure 1C*). In contrast, replacing the TCA codons with either TCC or TCT had little or no effect on colony morphology whereas switching to TCG increased the wrinkled appearance of the colonies (*Figure 1C*), which is indicative of robust biofilm formation.

Next, we asked whether switching serine codons was altering the level of SinR protein in liquid biofilm-inducing medium. Immunoblot analysis with anti-SinR antibodies revealed slightly yet consistently higher SinR levels in the strain with the AGT variant of *sinR* when compared to either the wild-type strain with three TCA codons or the TCG variant of *sinR* (*Figure 1D,E*).

SinR is highly similar (85% identity) to SlrR, which also plays a critical role in biofilm formation and whose gene (*slrR*) is under the direct negative control of SinR (*Chu et al., 2008*). Because SlrR cross reacts with the anti-SinR antibodies, we were also able to detect SlrR in our immunoblot analysis. Strikingly, the levels of SlrR were almost perfectly anti-correlated with those of SinR, with the differences in the SlrR protein levels among the *sinR* synonymous variants being much higher than the corresponding differences in SinR protein levels (*Figure 1F*). Because repression by SinR is ultrasensitive to SinR levels (*Chai et al., 2011*), small differences in SinR protein levels among *sinR* synonymous variants might be sufficient to cause large differences in the levels of expression of SinR-repressed genes such as *slrR*. Consistent with this idea, an *eps-lacZ* transcriptional fusion reporter for the SinR-repressed *epsA-O* matrix operon showed that the four TCN *sinR* variants had 3- to 19-fold higher *β*-galactosidase activity than the AGT and AGC variants (*Figure 1G*).

To test the generality of the observed hierarchy between the synonymous variants of *sinR*, we generated an additional set of eleven *sinR* synonymous variants in which we replaced either three TCC codons or two AGC/AGT codons (*Figure 1—figure supplement 1*) with their synonymous counterparts. Eight of these variants conformed to the hierarchy described above, namely, the four TCN variants behaved oppositely to the two AGC/AGT variants in colony morphology and in *eps-lacZ* reporter expression (*Figure 1—figure supplement 2*). The three variants that did not conform to the hierarchy could potentially reflect alterations to the mRNA sequence context near the mutation rather than the effect of a synonymous substitution per se. Taken together, the above results suggest that serine synonymous codons in the *sinR* coding sequence have a stereotypical effect on biofilm formation that is primarily determined by the differential usage of the four TCN and the two AGC/AGT codons.

## Entry into biofilm formation is accompanied by codon-specific increase in ribosome density

What is the mechanism by which serine codon usage affects SinR protein levels and biofilm formation? Synonymous codon changes can alter the synthesis of the encoded protein through changes in the

translation initiation rate, mRNA levels or the ribosome elongation rate (*Plotkin and Kudla, 2010*). However, the effect of synonymous codon usage on the initiation rate and mRNA levels is context-specific; only codons near the AUG start site affect translation initiation (*Kudla et al., 2009*), whereas only codons that are located in certain regions of secondary structure or at ribonuclease cleavage sites affect mRNA levels (*Bernstein et al., 2002*). Our observation that synonymous codon replacements at multiple locations along *sinR* have a stereotypical effect on biofilm formation argues against such context-specific mechanisms (except for the three exceptional cases noted above).

To test the alternative hypothesis that serine codon usage might alter the ribosome elongation rate, and given that the ribosome elongation rate at a codon varies inversely with the average ribosome density at that codon, we measured ribosome density on mRNAs at single-codon resolution using the ribosome profiling method (*Ingolia et al., 2009*; *Oh et al., 2011*; *Ingolia et al., 2012*). We grew *B. subtilis* in liquid biofilm-inducing medium, harvested cells either during exponential phase growth ($OD_{600}$ = 0.6) or during stationary phase when biofilm formation is induced ($OD_{600}$ = 1.4), and performed deep-sequencing of ribosome protected mRNA fragments and size-matched total mRNA fragments. Ribosome profiling yielded 3.75 and 2.55 million sequencing reads aligning to annotated protein-coding sequences for the exponential phase sample and the biofilm entry sample, respectively. The number of reads aligning to a single codon on individual mRNAs was too low for accurate quantification of ribosome density, and was not sufficient to directly detect increased ribosome density on the *sinR* transcript. However, we reasoned that the global pattern of ribosome density at codons across all mRNAs should reflect the ribosome density on individual transcripts such as *sinR*. Therefore, we calculated the median ribosome density at each of the 61 sense codons across the 1556 protein-coding sequences in the exponential phase sample and the 1148 sequences in the biofilm entry sample that had an average coverage greater than one sequencing read per codon (*Figure 2—figure supplement 1*, 'Materials and methods'). This analysis reproduced the previous observation (*Li et al., 2012*) of increased ribosome density 8 to 11 nt downstream of Shine-Dalgarno-like trinucleotide sequences both during exponential phase and during biofilm entry (*Figure 2—figure supplement 2*).

During exponential phase, median ribosome density varied over a twofold range, with no systematic difference between serine codons and the remaining codons (*Figure 2A*). By contrast, during biofilm entry, median ribosome density was significantly higher at serine and cysteine codons as compared to the remaining codons (*Figure 2B*), suggesting that the translation speed of ribosomes is selectively reduced at these codons during biofilm entry. Notably, the ribosome density at serine codons was not uniform: the four UCN codons had 1.9 to 2.1-fold higher ribosome density whereas the AGC and AGU codons had only 1.1 and 1.3-fold higher ribosome density respectively, relative to the median value across 61 sense codons. Further, this difference in ribosome density between UCN codons and the AGC/AGU codons was essentially identical when computed separately for codons located in the first half or in the second half of each gene (*Figure 2—figure supplement 3*), a finding that underscores the statistical robustness and the context independence of the observed difference.

## Serine starvation reduces translation speed and inhibits SinR synthesis in a codon-specific manner

Because ribosomes density increased at serine and cysteine codons only during biofilm formation and not during exponential phase growth, we hypothesized that the increased ribosome density was caused by the depletion of intracellular pools of these two amino acids during biofilm entry rather than by any intrinsic feature of the mRNA (*Li et al., 2012*) or the nascent polypeptide (*Charneski and Hurst, 2013*). Our hypothesis is also supported by the previous observation that synonymous codon usage can have a starvation-specific effect on protein levels (*Subramaniam et al., 2013*). Further, serine is a precursor metabolite for the biosynthesis of cysteine (*Gagnon et al., 1994*); hence cysteine depletion was likely the result of a decrease in intracellular serine concentration. Consistent with this hypothesis, we also observed an increase in ribosome density at both serine and cysteine codons during serine starvation of a *B. subtilis* serine-auxotrophic mutant (*Figure 3C*). Importantly, serine starvation resulted in an increase in ribosome density at only the four UCN serine codons but not at the AGC and AGU codons, matching the hierarchy seen during biofilm entry (*Figure 2B*). Serine starvation also resulted in differential levels of production of SinR-YFP protein fusions bearing different synonymous serine codons, whereas serine-rich growth resulted in identical levels of fusion protein production from these variants (*Figure 3A,B*). Finally, the addition of excess serine or cysteine (but not

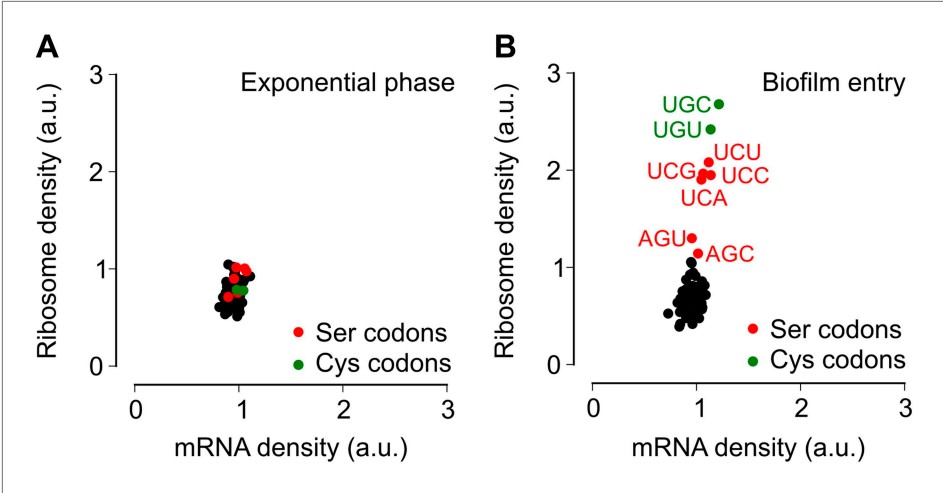

**Figure 2**. Entry into biofilm formation is accompanied by codon-specific increase in ribosome density. Genome-wide median ribosome density and total mRNA density at 61 sense codons (excluding start and stop codons) (**A**) during exponential phase growth (OD$_{600}$ = 0.6), and (**B**) during stationary phase when biofilm formation is induced (OD$_{600}$ = 1.4). The six serine (red) and two cysteine (green) codons are highlighted. Genome-wide ribosome density and total mRNA density were measured by deep-sequencing of ribosome-protected mRNA fragments and total mRNA fragments respectively, of a *B. subtilis* 3610 derivative (Δ*epsH*) grown in liquid biofilm-inducing medium.

The following figure supplements are available for figure 2:

**Figure supplement 1**. Computational workflow for deep-sequencing data analysis.

**Figure supplement 2**. Increase in ribosome density downstream of Shine-Dalgarno-like trinucleotide sequences.

**Figure supplement 3**. Context independence of ribosome and mRNA densities during biofilm formation.

any of the other 18 amino acids) blocked biofilm formation in wild type cells as judged after 48 hr of growth in biofilm-inducing medium (*Figure 3—figure supplement 1*, and data not shown).

Given that both biofilm entry and serine starvation resulted in increased ribosome density at serine and cysteine codons, we asked whether these two apparently different conditions invoke the same gene expression program in *B. subtilis*. The fold-change in average ribosome density of *B. subtilis* genes was positively correlated between biofilm entry and serine starvation (*Figure 3D*, $R^2$ = 0.27, p=$10^{-5}$, 1724 genes). However, a subset of 68 genes was induced at least 10-fold higher upon biofilm entry than during serine starvation (indicated by red markers in *Figure 3D*, *Table 1*). This subset included genes for anaerobic metabolism such as *lctEP*, *nasDE*, and *cydAB*, and is consistent with the recently proposed role of impaired respiration in biofilm formation (*Kolodkin-Gal et al., 2013*). We observed that sulfur metabolism genes were also enriched in this subset, possibly indicating a stronger response to cysteine depletion during biofilm entry than during serine starvation.

*In toto*, the results with the serine auxotroph support the inference that ribosome stalling observed during biofilm formation is due to a drop in intracellular serine levels. Our efforts to measure intracellular serine levels directly during growth in minimal, biofilm-inducing medium (MSgg) have been unsuccessful. Hence, we cannot rule out the less likely possibilities that biofilm entry causes serine and cysteine to be sequestered away from protein synthesis or that aminoacylation rate of the corresponding tRNAs decreases without changes in the intracellular serine and cysteine pools.

## Serine codon bias in biofilm-regulated genes reflects their expression during serine starvation

Here we found that the translation speed of ribosomes decreases at UCN serine codons and thereby modulates production of SinR and entry into biofilm formation. Based on our genome-wide measurements of ribosome density, we expect that the translation speed of ribosomes should decrease during biofilm entry not only on the *sinR* mRNA, but also on other mRNAs that are enriched for any of the four

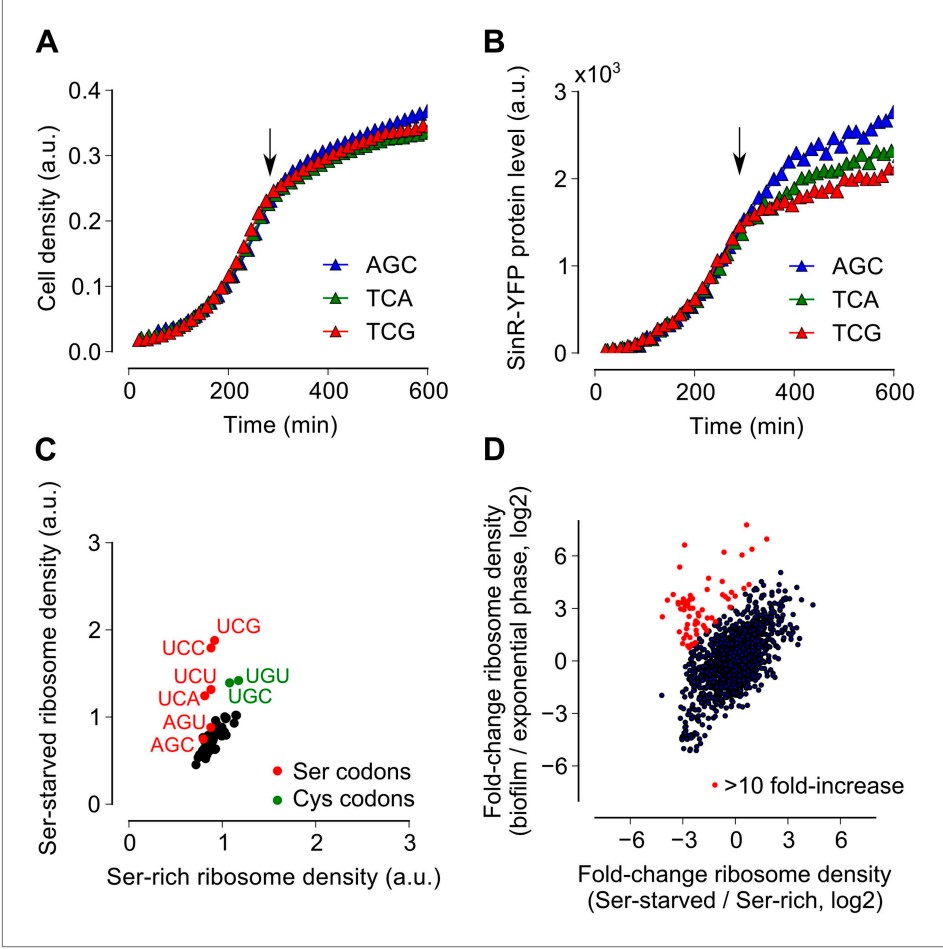

**Figure 3**. Serine starvation reduces translation speed and inhibits SinR synthesis in a codon-specific manner. (**A** and **B**) Three *sinR* synonymous variants were synthesized with 10 serine codons switched to AGC, TCA or TCG. The variants were expressed as SinR-YFP fusions from the *amyE* locus under the control of a *lac* promoter in a 3610-Δ*serA* serine auxotroph strain growing in serine-limited medium. Black arrow around 300 min indicates the onset of serine starvation caused by depletion of exogenously-added serine in the growth medium. Cell density (**A**) and the corresponding SinR-YFP protein level (**B**) were monitored using a 96-well spectrophotometer. (**C**) Genome-wide median ribosome density for 61 sense codons (excluding start and stop codons) during serine starvation (vertical axis) and serine-rich growth (horizontal axis) of a serine auxotrophic strain. (**D**) Fold-change in average ribosome density for individual genes upon biofilm entry (vertical axis) or serine starvation (horizontal axis). Genes that were preferentially up-regulated at least 10-fold upon biofilm entry in comparison to serine starvation are highlighted in red (68 genes, *Table 1*). Only genes with a minimum of 100 ribosome profiling reads in at least one of the samples were included in this analysis (1724 genes) and the reported log2 fold-changes are median-subtracted values across this gene set.

The following figure supplements are available for figure 3:

**Figure supplement 1**. Addition of excess serine or cysteine blocks pellicle formation by *B. subtilis*.

UCN codons. For example, we found that the four TCN codons are over-represented in two nucleotide biosynthesis genes, *pyrAA* and *purB* (**Figure 4A**). These two genes are also transcriptionally down regulated during biofilm entry (**Figure 4B**). A high frequency of TCN codons in these genes might serve to reinforce their transcriptional down regulation by reducing translation speed. Conversely, TCN codons are under-represented in the genes encoding lactate dehydrogenase, *ldh* (**Cruz Ramos et al., 2000**) and a master regulator of post-exponential phase gene expression, *spo0A* (**Molle et al., 2003**) (**Figure 4A**). The two genes are transcriptionally up regulated upon biofilm entry (**Figure 4B**). The low frequency of TCN codons in these genes might represent a mechanism for optimizing the production of their protein products by minimizing the slowing down of translation during biofilm entry.

**Table 1.** *B. subtilis* genes that have greater than 10-fold difference in expression ratio between biofilm formation and serine starvation

| Gene | A | B | C | D | E | F | Function |
|------|------|-------|--------|---------|--------|------|----------|
| albA | 9.35 | −0.69 | 32 | 10129 | 175 | 144 | antilisterial bacteriocin (subtilosin) production protein |
| alsD | 6.38 | 0.95 | 34 | 1384 | 53 | 135 | alpha-acetolactate decarboxylase |
| alsS | 6.96 | 1.79 | 76 | 4613 | 86 | 396 | acetolactate synthase |
| cah | 2.08 | −2.74 | 2531 | 5254 | 1486 | 295 | S-deacylase |
| ctc | 3.78 | −0.73 | 327 | 2206 | 322 | 257 | 50S ribosomal protein L25 |
| cydA | 5.36 | −3.18 | 234 | 4720 | 408 | 60 | cytochrome bd ubiquinol oxidase subunit I |
| cydB | 9.6 | −1.88 | 9 | 3374 | 129 | 46 | cytochrome bd ubiquinol oxidase subunit II |
| cysK | 1.52 | −2.21 | 15760 | 22120 | 8991 | 2580 | cysteine synthase |
| gcvPA | 0.97 | −2.38 | 1500 | 1440 | 1001 | 255 | glycine dehydrogenase subunit 1 |
| gcvPB | 1.05 | −2.28 | 2029 | 2057 | 1224 | 335 | glycine dehydrogenase subunit 2 |
| gspA | 4.54 | −0.75 | 111 | 1261 | 135 | 106 | glycosyl transferase (general stress protein) |
| iseA | 2.24 | −1.14 | 1117 | 2579 | 2521 | 1516 | inhibitor of cell-separation enzymes |
| lctP | 6.62 | −2.9 | 62 | 3009 | 247 | 44 | L-lactate permease |
| ldh | 3.47 | −3.88 | 2810 | 15,254 | 1732 | 157 | L-lactate dehydrogenase |
| maeN | 4.08 | −1.61 | 148 | 1226 | 347 | 151 | Na+/malate symporter |
| mccA | 3.23 | −2.72 | 759 | 3494 | 386 | 78 | cystathionine beta-synthase |
| metE | 2.72 | −2.13 | 33,027 | 106473 | 32,231 | 9760 | 5-methyltetrahydropteroyltriglutamate/homocysteine S-methyltransferase |
| mgsR | 4.72 | −1.53 | 163 | 2099 | 202 | 93 | stress transcriptional regulator |
| mtlA | 3.92 | −0.22 | 126 | 934 | 104 | 119 | PTS system mannitol-specific transporter subunit IICB |
| mtnA | 1.84 | −2.79 | 2522 | 4425 | 3501 | 671 | methylthioribose-1-phosphate isomerase |
| mtnD | 1.72 | −1.63 | 4095 | 6600 | 3182 | 1361 | acireductone dioxygenase |
| mtnK | 2.54 | −2.64 | 4290 | 12,196 | 5866 | 1250 | methylthioribose kinase |
| nasD | 6.21 | −0.65 | 217 | 7872 | 635 | 536 | assimilatory nitrite reductase subunit |
| nasE | 6.05 | 0.38 | 34 | 1106 | 63 | 109 | assimilatory nitrite reductase subunit |
| rbfK | 3.74 | −2.57 | 1329 | 8681 | 434 | 97 | RNA-binding riboflavin kinase |
| sboA | 7.77 | 0.64 | 121 | 12,895 | 256 | 530 | subtilosin A |
| sboX | 8.26 | 0.19 | 21 | 3128 | 76 | 115 | bacteriocin-like product |
| ssuA | 2.09 | −2.3 | 3691 | 7682 | 877 | 236 | aliphatic sulfonate ABC transporter binding lipoprotein |
| ssuB | 2.57 | −2.09 | 2416 | 7004 | 779 | 242 | aliphatic sulfonate ABC transporter ATP-binding protein |
| ssuC | 2.17 | −2.16 | 3784 | 8362 | 692 | 206 | aliphatic sulfonate ABC transporter permease |
| ssuD | 2.2 | −2.19 | 12,961 | 29,135 | 2812 | 817 | alkanesulfonate monooxygenase |
| tcyJ | 3.25 | −3.26 | 1046 | 4856 | 427 | 59 | sulfur-containing amino acid ABC transporter binding lipoprotein |
| tcyK | 3.79 | −3.55 | 2815 | 19,087 | 1095 | 124 | sulfur-containing amino acid ABC transporter binding lipoprotein |
| tcyL | 3.33 | −3.05 | 855 | 4223 | 387 | 62 | sulfur-containing amino acid ABC transporter permease |
| tcyM | 3.54 | −2.97 | 1859 | 10,556 | 581 | 99 | sulfur-containing amino acid ABC transporter permease |
| tcyN | 3.38 | −2.74 | 3363 | 17,149 | 1119 | 223 | sulfur-containing amino acid ABC transporter ATP-binding protein |
| ureA | 4.36 | 0.78 | 124 | 1252 | 77 | 176 | urease subunit gamma |
| ycgL | 0.99 | −3.01 | 512 | 500 | 278 | 46 | hypothetical protein |
| ycgM | 2.46 | −1.35 | 39 | 105 | 178 | 93 | proline oxidase |
| ycgN | 2.45 | −1.6 | 928 | 2475 | 1267 | 556 | 1-pyrroline-5-carboxylate dehydrogenase |
| ycnJ | 0.75 | −2.63 | 168 | 138 | 121 | 26 | copper import protein |
| ydaG | 4.14 | 0.49 | 70 | 604 | 80 | 150 | general stress protein |
| ydbL | 3.07 | −0.31 | 294 | 1210 | 130 | 139 | hypothetical protein |

*Table 1. Continued on next page*

Table 1. Continued

| Gene | A | B | C | D | E | F | Function |
|------|------|------|------|------|------|------|----------|
| yeaA | 1.34 | −2.59 | 112 | 138 | 139 | 31 | hypothetical protein |
| yezD | 2.52 | −4.17 | 145 | 406 | 340 | 25 | hypothetical protein |
| yitJ | 3.01 | −2.43 | 2785 | 10,990 | 4687 | 1153 | bifunctional homocysteine S-methyltransferase/ 5,10-methylenetetrahydrofolate reductase |
| yjbC | 3.67 | −0.57 | 104 | 647 | 248 | 222 | thiol oxidation management factor; acetyltransferase |
| yjnA | 1.5 | −2.81 | 972 | 1342 | 790 | 149 | hypothetical protein |
| yoaB | 2.28 | −2.26 | 2206 | 5244 | 2862 | 792 | negatively charged metabolite transporter |
| yoaC | 2.92 | −1.89 | 1200 | 4459 | 1436 | 513 | hydroxylated metabolite kinase |
| yrhB | 2.92 | −2.93 | 3806 | 14,096 | 1904 | 333 | cystathionine beta-lyase |
| yrrT | 2.97 | −3.12 | 546 | 2089 | 443 | 68 | AdoMet-dependent methyltransferase |
| ytlI | 1.66 | −3.2 | 206 | 318 | 141 | 20 | LysR family transcriptional regulator |
| ytmI | 3.34 | −3.27 | 3452 | 17,173 | 1640 | 226 | N-acetyltransferase |
| ytmO | 3.4 | −2.88 | 3866 | 20,007 | 1179 | 213 | monooxygenase |
| ytnI | 3 | −2.6 | 3522 | 13,770 | 867 | 189 | redoxin |
| ytnJ | 3.14 | −2.86 | 10,645 | 45,997 | 2495 | 456 | monooxygenase |
| ytnL | 3.56 | −2.45 | 1281 | 7371 | 354 | 86 | aminohydrolase |
| ytnM | 3.5 | −2.6 | 4542 | 25,202 | 1264 | 277 | transporter |
| yuaF | 1.89 | −1.74 | 87 | 157 | 158 | 63 | membrane integrity integral inner membrane protein |
| yvzB | 0.95 | −2.48 | 125 | 118 | 168 | 40 | flagellin |
| yxaL | 3.54 | −0.41 | 1069 | 6077 | 442 | 442 | membrane associated protein kinase |
| yxbB | 3.7 | −0.01 | 108 | 685 | 118 | 155 | S-adenosylmethionine-dependent methyltransferase |
| yxeK | 0.86 | −2.72 | 2702 | 2406 | 1073 | 216 | monooxygenase |
| yxeL | 1.29 | −2.94 | 437 | 525 | 202 | 35 | acetyltransferase |
| yxeM | 0.87 | −2.57 | 3003 | 2692 | 1047 | 233 | ABC transporter binding lipoprotein |
| yxeP | 1.75 | −2.24 | 2577 | 4246 | 736 | 207 | amidohydrolase |
| yxjH | 2.02 | −1.82 | 4162 | 8243 | 3811 | 1432 | methyl-tetrahydrofolate methyltransferase |

A—median-subtracted log2 fold-change: biofilm/exponential-phase, B—median-subtracted log2 fold-change: serine starvation/serine rich, C—raw counts: biofilm entry, D—raw counts: exponential phase, E—raw counts: serine rich, F—raw counts: serine starvation.

Consistent with this idea, replacement of AGC/AGT codons by TCN codons in *spo0A*, whose protein product positively regulates biofilm entry (by turning on the synthesis of the SinR antagonist SinI), resulted in defective biofilm formation (*Figure 4C*) in contrast to the stimulatory effect of replacing AGC/AGT codons with TCN codons in *sinR* (*Figure 1—figure supplement 2*).

## Discussion

Together, our results implicate serine depletion as an environmental cue that contributes to promoting biofilm formation in *B. subtilis* together with other cues that are sensed by the histidine kinases KinA–D. Serine depletion is sensed through a remarkably simple mechanism based on reduced translation speed at UCN serine codons in the mRNA for a regulatory protein, SinR, whose repressive effects are highly sensitive to small changes in the level of the protein. We presume that UCN codons lower SinR levels simply by slowing the rate of ribosome movement along the mRNA (elongation). However, it is possible that the reduced translation speed at UCN codons during biofilm entry could be followed by downstream events such as ribosome rescue (*Keiler et al., 1996*) or mRNA decay (*Hayes and Sauer, 2003*) that might also contribute to lowering the levels of the SinR protein.

The serine sensing mechanism uncovered here operates through over-representation of the TCN serine codons in the *sinR* gene without the necessity for any other dedicated protein or RNA for sensing serine depletion. By contrast, transcriptional attenuation, a widespread mechanism among bacteria

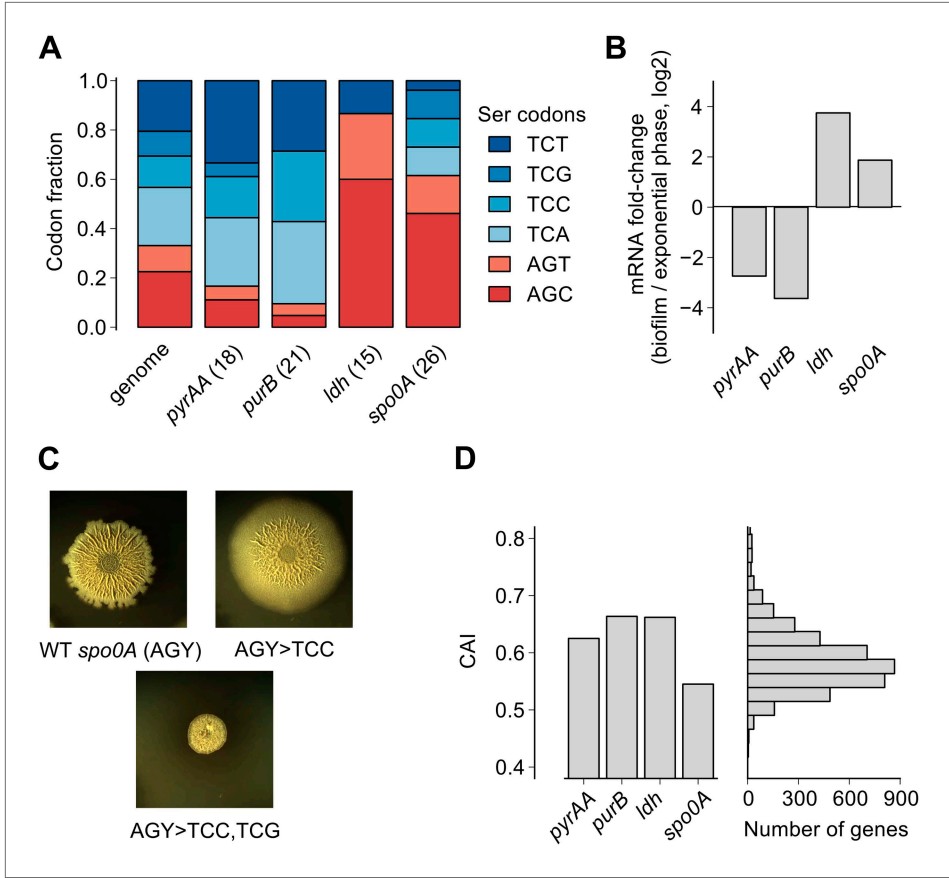

**Figure 4**. Serine codon bias of biofilm-regulated genes reflects their expression under serine starvation. (**A**) Relative serine codon fraction in genes for nucleotide biosynthesis (*pyrAA*, *purB*), lactate dehydrogenase (*ldh*) and a sporulation regulator (*spo0A*). Numbers in parentheses indicate the number of serine codons in each gene. Relative fraction of serine codons across the *B. subtilis* genome is shown for comparison. (**B**) Fold-change (expressed in log2 units) in average ribosome density upon biofilm entry for the four genes shown in **A**. (**C**) Colony morphology of a wild-type strain and two *spo0A* synonymous variants grown on solid biofilm-inducing medium. Seven AGC/AGT codons in wild-type *spo0A* were replaced by either 7 TCC codons or 3 TCC and 4 TCG codons and inserted at the chromosomal *spo0A* locus. Both the wild-type *spo0A* and the synonymous *spo0A* variants were inserted with a downstream selection marker. (**D**) Left: Codon Adaptation Index (CAI) for the four genes shown in **A**. Right: Distribution of CAI values for 4153 protein-coding sequences of *B. subtilis*.

for sensing amino acids that also relies on changes in translation speed, involves a translation-transcription coupling mechanism that is mediated by highly structured mRNAs and leader peptides (**Henkin and Yanofsky, 2002**).

We note that the biased usage of the four TCN serine codons, which act as starvation sensors during biofilm formation, is not evident from widely-used phenomenological measures of codon bias such as the codon adaptation index (**Figure 4D**), which primarily reflects codon preferences during exponential growth (**Sharp and Li, 1987**; **Andersson and Kurland, 1990**). The difference in translation speed between the four UCN codons and the two AGC/AGU codons under biofilm-inducing conditions is likely mediated by differences in concentration of the corresponding aminoacylated tRNAs (**Elf et al., 2003**; **Dittmar et al., 2005**), as was recently observed in serine-starved *E. coli* (**Subramaniam et al., 2013**). Interestingly, the hierarchy between UCN codons and AGC/AGU codons in *B. subtilis* during serine starvation is similar to the one in *E. coli* even though copy numbers of the corresponding tRNA genes have diverged significantly between these two organisms (**Lowe and Eddy, 1997**). Despite different tRNA gene copy numbers, it is possible that the relative abundances of the serine tRNA isoacceptors are similar between the two organisms or that their relative abundances might be regulated in the same manner in response to nutrient deprivation (**Doi et al., 1966**).

Serine is one of the first amino acids to be completely consumed from the culture medium when either *B. subtilis* or *E. coli* cells are grown in complex rich medium (*Liebs et al., 1988*; *Prüss et al., 1994*; *Sezanov et al., 2007*). Indeed, increased ribosome density has been observed at serine codons during growth of *E. coli* in Luria-Bertani broth (*Li et al., 2012*). Thus the role of synonymous serine codons as starvation sensors discovered here in the specific context of biofilm formation might be a general regulatory strategy in microbes for adapting to nutrient depletion at the end of exponential phase growth. It is noteworthy that depletion of specific amino acids affects developmental transitions in several eukaryotic cells (*Marin, 1976*; *Sundrud et al., 2009*; *Wang et al., 2009*). It will be interesting to test whether a codon-based sensing mechanism, similar to the one found here in bacterial biofilm development, also plays a role in eukaryotic cells during amino acid depletion.

## Materials and methods

### Bacterial strains and media

For ribosome profiling during biofilm formation, a 3610-Δ*epsH* strain (RL3852) was used to ensure dispersed growth in liquid media (*Kearns et al., 2005*). For serine starvation experiments, a serine-auxotrophic 3610-Δ*serA* strain (YC865) was used. A list of strains, plasmids, and oligonucleotides used in this work are summarized in *Supplementary file 1*.

For general purposes, *B. subtilis* strains PY79, 3610, and their derivatives were grown in Luria-Bertani (LB) medium (10 g tryptone, 5 g yeast extract, and 5 g NaCl per liter broth) at 37°C. Antibiotics were added to the media at the following concentrations for *B. subtilis* strains: 10 µg ml⁻¹ of tetracycline, 100 µg ml⁻¹ of spectinomycin, 10 µg ml⁻¹ of kanamycin, 5 µg ml⁻¹ of chloramphenicol, and 1 µg ml⁻¹ of erythromycin. Minimal MSgg medium was used as the liquid biofilm-inducing medium. The same medium with 1.5% Bacto-agar (Difco, Franklin Lakes, NJ) was used as the solid biofilm-inducing medium. MSgg medium composition: 5 mM potassium phosphate (pH 7), 100 mM MOPS (pH 7), 2 mM $MgCl_2$, 700 µM $CaCl_2$, 50 µM $MnCl_2$, 50 µM $FeCl_3$, 1 µM $ZnCl_2$, 2 µM thiamine, 0.5% glycerol, 0.5% glutamate, 50 µg ml⁻¹ tryptophan, 50 µg ml⁻¹ phenylalanine and 50 µg ml⁻¹ threonine. For overnight growth of serine auxo-trophic 3610 strains, MSgg medium was supplemented with serine to a final concentration of 5 mM. For serine starvation experiments in which YFP fluorescence was measured (*Figure 3A,B*), MSgg medium was supplemented with 800 µM serine and 400 µM serine methyl-ester (Sigma, St. Louis, MO). Serine methyl-ester was added to ensure slow growth under serine starvation conditions.

### Strain construction

General methods for molecular cloning followed published protocols (*Sambrook 2001*). SPP1 phage-mediated transduction was used to transfer antibiotic-marked DNA fragments between different strains (*Kearns et al., 2005*). Long-flanking PCR mutagenesis was applied to generate insertional deletion mutations (*Wach 1996*). Synonymous switches in *sinR* and *spo0A* were generated by using synthetic DNA fragments (Genewiz, South Plainfield, NJ) or by applying site-directed mutagenesis (Roche, Switzerland). Sequences of the primers used in constructing mutant *sinR* alleles are described in *Supplementary file 1*. Incorporation of synonymous substitutions into the *sinR* or *spo0A* gene at the native locus was done by allele exchange and followed a method described previously (*Chai et al., 2011*).

### Biofilm assays

*B. subtilis* cells were first grown in LB broth at 37°C to mid-exponential phase. For formation of biofilm colonies, 2 µl of the cells was spotted onto MSgg medium solidified with 1.5% agar. Plates were incubated at 23°C for 3–4 days before analysis. All images were taken using either a Nikon CoolPix 950 digital camera or using a SPOT camera (Diagnostic Instruments, Sterling Heights, MI). Assays for the β-galactosidase activities were described previously (*Kearns et al., 2005*).

### Genetic screen for suppressor mutants

Following a previously-published protocol (*Chai et al., 2010*), we set up a genetic screen to search for spontaneous mutations that suppressed the defective biofilm phenotype of a *B. subtilis* Δ*slrR* mutant (YC131). The defective biofilm phenotype of the Δ*slrR* mutant is manifested as an inability to form robust floating pellicles (*Chu et al., 2008*; *Chai et al., 2010*). Briefly, the Δ*slrR* strain was inoculated into liquid MSgg medium in 6-well plates and incubated at 30°C. After 48 hr, pellicle formation was examined visually. In some wells, robust pellicles appeared possibly due to a second, suppressor mutation elsewhere in the genome. Cells from those wells were picked and streaked out on fresh LB

agar plates to isolate single colonies. Cells from the single colonies were then tested for altered colony morphology on solid MSgg medium to confirm the suppressor phenotype. Similar genetic screens in previous studies (*Kearns et al., 2005*; *Chai et al., 2010*) had established that mutations in the *sinR* gene could suppress the defective biofilm phenotype of the Δ*slrR* mutant. Therefore, we isolated genomic DNA from the putative suppressor mutants, amplified the *sinR* gene by PCR, and then sequenced the *sinR* locus. Once a mutation in the *sinR* gene was confirmed by sequencing, the same mutation was reconstituted in the wild type background (3610) following a previous protocol (*Chai et al., 2010*), and assayed for alteration in colony morphology on solid MSgg medium.

## Bacterial growth for ribosome profiling

For ribosome profiling during biofilm formation, fresh colonies were inoculated into 8 ml of MSgg liquid medium and grown for 12 hr at 30°C, 200 rpm. Saturated cultures were diluted 1:1000 into 200 ml aliquots of fresh MSgg medium and shaken in a 1L flask at 30°C, 200 rpm. For exponential-phase ribosome profiling (*Figure 2A*), cultures were harvested at $OD_{600} = 0.6$. For ribosome profiling during biofilm entry (*Figure 2B*), cultures were harvested at $OD_{600} = 1.4$. For the serine repletion experiment (*Figure 4*), serine was added to a final concentration of 2.5 mM at $OD_{600} = 1.4$ and harvested after 30 min at 30°C, 200 rpm. For the serine starvation experiment (*Figure 3B,C*), pre-cultures were grown in MSgg medium supplemented with 5 mM serine and then diluted into 200 ml of the same medium. At an $OD_{600} = 0.6$, the cultures were filtered and re-suspended either in MSgg medium (starvation) or in MSgg medium with 5 mM serine (control), and harvested after 60 min at 30°C, 200 rpm (*Figure 3D*).

## Western blotting

Cultures were grown with shaking at 37°C, and 14 ml culture aliquots were harvested at an $OD_{600}$ between 2.0 and 2.5. Cell pellets were collected by centrifugation and washed once with 10 ml of lysis buffer (20 mM Tris-HCl, 200 mM NaCl, 1 mM EDTA pH 7.4). Pellets were resuspended in 1.2 ml lysis buffer and incubated with 20 µg/ml of lysozyme (Sigma) for 1 hr on ice. Cells were further lysed by sonication. Cell debris was removed by centrifugation (14,000 rpm, 30 min, 4°C). The concentration of total protein in the lysates was determined by a Bradford assay (Bio-Rad, Hercules, CA). Samples for SDS-PAGE were prepared in Laemmli buffer normalized to equal protein concentration. Samples were ran on an NuPAGE 12% gel (Invitrogen, Carlsbad, CA, 1.0 mm, Bis/Tris, 200 V, ~50 min) and transferred to a PVDF membrane (Millipore, Billerica, MA) at 100 V for 1 hr. The membrane was blocked in 5% milk-TPBS for 1 hr, and then incubated with anti-SinR antibody (1:2500, polyclonal) and anti-SigA (1:100,000, polyclonal) overnight. The membranes were washed 3 times in TPBS for 5 min each. Blots were incubated with goat anti-rabbit secondary antibody conjugated to Horseradish peroxidase (1:10,000, Bio-Rad). Blots were washed three times in TPBS, and developed with SuperSignal West Dura chemiluminescent substrate (Thermo, Waltham, MA) and imaged on a gel-doc (Bio-Rad). Densitometry analysis of Western blot images was performed using ImageJ software (NIH, http://rsbweb.nih.gov/ij/). Rectangles were drawn around each distinct band and the average pixel intensity in this rectangle was calculated, followed by subtraction of background pixel intensity from an identical rectangle drawn in a region without any band. These background subtracted values were used for calculating SinR/SigA, SlrR/SigA and SinR / SlrR ratios in *Figure 1*.

## YFP fluorescence measurements

A previously published measurement protocol was used (*Subramaniam et al., 2013*). Fresh colonies were inoculated into 1 ml of MSgg liquid medium with 5 mM serine and grown overnight in deep 96-well plates at 30°C, 1400 rpm. In the morning, saturated cultures were diluted 1:100 into 1 ml of MSgg medium with 800 µM serine and 400 µM serine methyl-ester and shaken at 30°C, 1350 rpm for 3 hr. Three aliquots of 150 µl from each culture was pipetted into three wells of three 96-well plates (3799, Costar, Corning, NY). Wallac Victor2 plate reader (PerkinElmer, Waltham, MA) was used to monitor cell density (absorbance at 600 nm) and YFP expression (fluorescence, excitation 504 nm and emission 540 nm). Each plate was read every 15 min using a robotic system (Caliper Life Sciences, Hopkinton, MA) and shaken at 1000 rpm in between readings (Variomag Teleshake shaker, Daytona Beach, FL). 30°C and 60% relative humidity was maintained throughout the experiment.

## Ribosome profiling and mRNA quantification

Ribosome profiling protocol was adapted from published literature (*Ingolia et al., 2009*; *Oh et al., 2011*; *Ingolia et al., 2012*) with minor modifications as described below. Briefly, 200 ml of bacterial

culture was harvested by filtration. The filter was immediately inserted into a 50 ml conical tube, flash frozen in liquid nitrogen, and stored at −80°C until further processing. Frozen cells were re-suspended in 8 ml of polysome resuspension buffer (20 mM Tris pH 8.0, 10 mM $MgCl_2$, 100 mM $NH_4Cl$, and 100 μg $ml^{-1}$ Chloramphenicol). Re-suspended cells were pelleted by centrifugation (3000 g, 4°C, 5 min) and the supernatant was discarded. The cell pellet was re-suspended in 500 μl of polysome lysis buffer (1X polysome resuspension buffer, 5 mM $CaCl_2$, 0.4% TritonX-100, 0.1% NP-40, and 100 U/ml RNase-free DNase [04716728001; Roche]), and transferred to an ice-cold 1.5 ml tube containing 500 μl of 0.2–0.3 μm acid-washed glass beads (G1277; Sigma). Cells were lysed by vortexing at maximum speed on a vortexer in a 4°C room (Vortex Genie 2, 10 × 30 s with 1 min cooling on ice in between). The lysate was clarified by centrifugation (20,000 g, 4°C, 10 min) and the supernatant was transferred to a fresh 1.5 ml tube. 500 μg of total RNA ($A_{260}$ units) was digested (25°C, 1400 rpm, 60 min, 150 μl vol) with 2 U/μg of Micrococcal nuclease (LS004797; Worthington, Lakewood, NJ). The digestion was quenched with 1.5 μl of 0.5 M EGTA, loaded on top of a 10–50% sucrose gradient and ultra-centrifuged in a SW41 rotor (35000 rpm, 4°C, 150 min). Monosomes collected by gradient fractionation (Biocomp Instruments, Canada).

For total RNA extraction, 100 μl of polysome lysate was mixed with 400 μl of RNA extraction buffer (0.3 M sodium acetate, 10 mM EDTA, pH 4.5) and the aqueous phase was extracted twice with phenol-chloroform and once with chloroform. RNA was precipitated with an equal volume of isopropanol. The pellet was washed with 70% ice-cold ethanol and re-suspended in 100 μl of 10 mM Tris pH 7.0. 10 μg of total RNA was DNase-treated and mRNA enriched using the Microbe Express kit (Invitrogen). mRNA was fragmented by heating at 95°C with a bicarbonate buffer (*Ingolia et al., 2009*) for 20 min.

Collected monosome fractions were purified using the same phenol-chloroform method as used for total RNA extraction above. Monosomes and fragmented mRNA were then used for small RNA sequencing library preparation. Size selection, dephosphorylation, polyadenylation, reverse-transcription, circularization and PCR amplification were performed using the same protocol as in (*Ingolia et al., 2009*). An rRNA subtraction step was carried out between the circularization and PCR amplification steps using the same protocol as in (*Oh et al., 2011*). Typically, several samples were multiplexed for sequencing in an Illumina HiSeq sequencer such that at least 1 million reads were obtained for each sample.

## Deep-sequencing data analysis

Deep-sequencing data analysis was carried out in Bash and Python programming languages, and performed on the Harvard research computing Odyssey cluster. Main steps are summarized below and shown schematically in *Figure 2—figure supplement 1*.

1. Each 50 nt single-end read was polyA-trimmed by identifying 10 or greater number of adjacent adenines and discarding all nucleotides starting from −1nt of the polyA run. The first 5′ nt of the read was also discarded since its identity was ambiguous in several reads. PolyA-trimmed reads were first aligned against all non-coding RNAs in the *B. subtilis* genome using bowtie aligner (ver. 0.12.7, *Langmead et al., 2009*).
2. The non-coding RNA sequences were downloaded from NCBI (NC_000964.frn).
3. Reads that did not align to non-coding RNAs were aligned against the whole *B. subtilis* genome using bowtie aligner. The *B. subtilis* genome was downloaded from NCBI (NC_000964.fna). Only reads that had less than three mismatches with the reference genome were considered for further analysis.
4. Reads that aligned to the *B. subtilis* genome were further trimmed by 8 nt from each end to approximate the ribosome A-site coordinate. The remaining sequence was normalized by its length and assigned to the corresponding genomic coordinate, and this value was designated as the read density at this genomic coordinate during further downstream analyses.
5. Average ribosome and mRNA density for a single gene was calculated by summing the read density between the start and stop codon, and then normalizing by the length of the gene.
6. Fold-change in ribosome density for a single gene between two samples was calculated by taking the log2 of the ratio of average ribosome density between the two samples for that gene. The median value of this log2 fold-change across all genes that received a minimum of 100 reads in at least one of the two samples was then subtracted from the fold-change value for each gene. This median-subtracted log2 fold change in reported throughout this work. We note that the average ribosome

density on any gene is directly proportional to the corresponding mRNA level in the absence of specific translational regulation. Hence fold-changes in ribosome density (such as the one shown in *Figure 3D*) primarily reflect fold-changes in mRNA level.

7. To calculate the ribosome and mRNA density at individual codons,

   a. The start codon was treated as a single separate codon irrespective of its identity.
   b. Only genes with average ribosome density of at least 1 read per codon were considered.
   c. The ribosome density at the first nucleotide of each codon was assigned as the ribosome density at that codon. For each of the 64 codons and the start codon, read density at the first nucleotide of the codon was averaged across all occurrences of that codon in a single gene and then normalized by the average ribosome density for that gene. Hence a codon without over- or under- representation of ribosome or mRNA density will have a density value equal to 1.
   d. Genome-wide ribosome and mRNA density was calculated as the median of the individual gene read density from Step (3) across all genes that pass the threshold of Step (2).
   e. Ribosome profiling measurement resulted in a high ribosome density at start and stop codons. It is unclear whether this increase is a true biological signal or caused by the measurement protocol (*Ingolia et al., 2011*). Hence these codons were excluded from the ribosome density plots shown in *Figure 2A,B and 3C*. However, the increased ribosome density at these codons resulted in a concomitant decrease in ribosome density at the remaining codons due to normalization by the average ribosome density for each gene (which included the start and stop codons).

## Codon usage analysis

Codon Adpatiaton Index (CAI) was calculated according to the original prescription of Sharp and Li (*Sharp and Li, 1987*). For this calculation, 68 genes that had an annotation as 'ribosomal' were used as the reference set of highly expressed genes. The annotations file used for this analysis was downloaded from NCBI (NC_000964.ptt). The p value for the higher frequency of TCN codons in *sinR* was calculated by assuming a binomial distribution of TCN and AGC/AGT codons. The p value represents the binomial probability that there are 10 or more TCN codons in the *sinR* gene (12 serine codons total) given the genome-wide frequency of serine codons (0.66 for TCN codons and 0.34 for AGC/AGT codons).

## Acknowledgements

We thank P Cluzel, AL Sonenshein, T Norman, and J Weissman for valuable comments during manuscript preparation, T Gao for performing some experiments, and B Burton for use of sucrose-gradient fractionator. Preliminary work by AS in the Cluzel lab was supported by a start-up grant from Harvard University. The computations in this paper were run on the Odyssey cluster supported by the FAS Science Division Research Computing Group at Harvard University.

## Additional information

### Competing interests

EO: Chief Scientific Officer and a Vice President at the Howard Hughes Medical Institute, one of the three founding funders of *eLife*. RL: Senior editor, *eLife*. The other authors declare that no competing interests exist.

### Funding

| Funder | Grant reference number | Author |
| --- | --- | --- |
| National Institutes of Health | GM18568 | Richard Losick |
| Howard Hughes Medical Institute | | Erin O'Shea |
| Northeastern University | | Yunrong Chai |
| National Institutes of Health | 1K99 GM107113 | Arvind R Subramaniam |

The funders had no role in study design, data collection and interpretation, or the decision to submit the work for publication.

## Author contributions

ARS, ADL, YC, Conception and design, Acquisition of data, Analysis and interpretation of data, Drafting or revising the article; NB, RL, Conception and design, Analysis and interpretation of data, Drafting or revising the article; YC, Conception and design, Acquisition of data, Analysis and interpretation of data; EO, Analysis and interpretation of data, Drafting or revising the article, Contributed unpublished essential data or reagents

## Additional files

### Supplementary files

• Supplementary file 1. Lists of strains, plasmids, and primers.

### Major dataset

The following dataset was generated:

| Author(s) | Year | Dataset title | Dataset ID and/or URL | Database, license, and accessibility information |
| --- | --- | --- | --- | --- |
| Subramaniam AR, DeLoughery A, Bradshaw N, Chen Y, O'Shea E, Losick R, Chai Y | 2013 | A Serine Sensor for Multicellularity in a Bacterium | GSE50870; http://www. ncbi.nlm.nih.gov/geo/ query/acc.cgi? acc=GSE50870 | Publicly available at NCBI GEO (http://www. ncbi.nlm.nih.gov/geo/). |

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
