## [Decision Letter]

Thank you for sending your work entitled “A Serine Sensor for Multicellularity in a Bacterium” for consideration at *eLife*. Your article has been favorably evaluated by two reviewers and one member of the Board of Reviewing Editors.

The Reviewing editor and the reviewers discussed their comments before we reached this decision, and the Reviewing editor has assembled the following comments to help you prepare a revised submission.

All agree that the study presents an exciting novel mechanism for the regulation of the transition to biofilm formation by *Bacillus subtilis*. In this mechanism, the cells sense the decrease in serine and cysteine abundance through changes in the speed of translation of genes that contain certain serine codons. It is noteworthy that these codons are not correlated with the usual codon usage, indicating a possible change in the tRNA levels or in the abundance of codons using them. The manuscript raises intriguing questions about the use of similar mechanisms by other organisms for developmental transitions or when cells are confronted with reductions in the availability of specific amino acids. However, the study would be strengthened if the authors addressed the following comments:

1) The conclusion that slow ribosome elongation at UCN codons reduces production of SinR protein to elicit biofilm formation is strongly implicated but not demonstrated. The authors should show ribosome profiles of genes with both types of codons, before and after Ser starvation or biofilm formation. The reviewers appreciate that expression of some genes might be low, but pausing in specific transcripts can be detected, and pausing at low abundance transcripts such as *sinR* should be detected by increasing sequencing coverage.

2) The data are largely compelling regarding the effect of different serine codons on SinR level. However the authors should discuss why certain codon substitutions behave in a non-stereotypical manner (Figure 1—figure supplement 2). While the hypothesis that ribosome pausing at these codons decreases translation elongation rate and thus SinR protein levels is the more intriguing possibility, an alternative hypothesis that pausing decreases *sinR* transcription via an attenuation-like mechanism has not been ruled out. The main analysis in Figure 1 involves replacing TCA codons with synonymous mutations. These three codons are toward the 5' end of the transcript where translation may affect transcription. The TCC codons are closer to the 3' end, and replacing them with AGC yields a result opposite to the one expected. Does surrounding mRNA sequence or structure have an impact?

3) The authors note that changes in expression of genes do not correlate with the codon usage of the specific Ser codons in ribosomal proteins. To understand how this is possible, the authors should determine if the codon usage in the total mRNA sample (based on the abundance of each codon in the total mRNA) changes during biofilm formation or during Ser starvation.

4) A major conclusion is that the UCN codons act as a sensor for serine starvation that is essential for biofilm formation. How is intracellular serine concentration regulated during biofilm formation? Do cells down-regulate serine biosynthesis during the transition to stationary phase? Do the authors see any transcriptional down-regulation of serine biosynthesis genes? This should be clarified. In addition, the authors mention that serine is among the first amino acids exhausted from the laboratory medium during exponential growth in complex medium. How does that apply to bacteria in non-laboratory conditions, where the presumed mechanism evolved?

---

## [Author Response]

*1) The conclusion that slow ribosome elongation at UCN codons reduces production of SinR protein to elicit biofilm formation is strongly implicated but not demonstrated. The authors should show ribosome profiles of genes with both types of codons, before and after Ser starvation or biofilm formation. The reviewers appreciate that expression of some genes might be low, but pausing in specific transcripts can be detected, and pausing at low abundance transcripts such as* sinR *should be detected by increasing sequencing coverage*.

We thank the reviewers for pointing that we have not explicitly demonstrated ribosome pausing on the *sinR* transcript. To avoid misinterpretation by the reader, we now clearly state this limitation in our revised text. Our ribosome profiling measurement did not allow us to robustly infer ribosome pausing at UCN codons on any individual transcript. At best, the selective ribosome pausing at UCN codons was evident only after calculating the median over the 10 transcripts with the highest ribosome density, while the median over the 3 transcripts with the highest ribosome density was significantly affected by noise (see Author response image 1 below). In our ribosome profiling measurement, averaging over the top 10 transcripts resulted in an average read density of ∼1500 reads per codon. By comparison, the average read density for the *sinR* transcript was ∼3 reads per codon. Therefore, we estimate that a ∼500X higher coverage (implying ∼2 billion total sequencing reads) is required to robustly detect ribosome pausing on the *sinR* transcript, which will be a prohibitively expensive experiment. Nevertheless, we are not aware of any experimental evidence or plausible mechanism by which the ribosome pausing at UCN codons that is inferred from a genome-wide median, is likely to be invalid at the level of individual mRNAs such as the *sinR* transcript. The magnitude of the specific ribosome pauses identified in Ref. 26 relative to the serine-starvation induced pauses observed here is uncertain, making a direct quantitative comparison difficult.Author response image 1.Sequencing read density around six serine codons, calculated as a median over the 3 genes with the highest ribosome density (left) or the 10 genes with the highest ribosome density (right). The sequencing read density for each gene was normalized by the total number of sequencing reads for that gene (same procedure as in Figure 2).

*2) The data are largely compelling regarding the effect of different serine codons on SinR level. However the authors should discuss why certain codon substitutions behave in a non-stereotypical manner (*Figure 1—figure supplement 2*). While the hypothesis that ribosome pausing at these codons decreases translation elongation rate and thus SinR protein levels is the more intriguing possibility, an alternative hypothesis that pausing decreases* sinR *transcription via an attenuation-like mechanism has not been ruled out. The main analysis in*
Figure 1
*involves replacing TCA codons with synonymous mutations. These three codons are toward the 5' end of the transcript where translation may affect transcription. The TCC codons are closer to the 3' end, and replacing them with AGC yields a result opposite to the one expected. Does surrounding mRNA sequence or structure have an impact*?

If an attenuation-like mechanism underlies the observed decrease in SinR protein levels due to UCN codons, such a mechanism is expected to also decrease levels of the *sinR* mRNA. On the contrary, in our genome-wide mRNA measurements, we observed that *sinR* mRNA levels increased slightly upon biofilm formation (median-normalized fold-change: 1.33). This result is also consistent with earlier observation that expression of *sinR* mRNA is almost constant [Gaur et al. J Bacteriol 170, 1046 (1988)]. Further, in addition to a ribosome pause site, transcriptional attenuation typically requires a leader peptide, and an anti-terminator RNA structure, neither of which are known to be present in the *sinR* transcript.

Introducing synonymous substitutions can affect expression levels of the encoded protein by several mechanisms including changes in mRNA stability or secondary structure. We agree with the reviewers that the local mRNA sequence is a possible determinant of the anomalous result observed with the TCC>AGC *sinR* variant as well as the two other *sinR* variants. We have now indicated this possibility in the main text. Based on our experience, such context-specific effects in a few variants are unavoidable when a large number of synonymous variants of a gene are systematically interrogated.

*3) The authors note that changes in expression of genes do not correlate with the codon usage of the specific Ser codons in ribosomal proteins. To understand how this is possible, the authors should determine if the codon usage in the total mRNA sample (based on the abundance of each codon in the total mRNA) changes during biofilm formation or during Ser starvation*.

Usage of the six serine codons in the total mRNA pool differed less than 20% between exponential phase and biofilm entry. Further, this change was not specific to the four serine-sensitive TCN codons. More generally, protein expression level does not correlate with codon usage even under nutrient-rich conditions. We expect that protein expression during amino acid limitation is determined by the aminoacylated fraction of tRNA isoacceptors cognate to the limiting amino acid. It is worth noting that the aminoacylated fraction of tRNA isoacceptors during amino acid limitation might even be inversely correlated with codon usage. The exact value of this aminoacylated fraction has been hypothesized to depend on the balance between supply and demand for aminoacylated tRNA isoacceptors during protein synthesis.

*4) A major conclusion is that the UCN codons act as a sensor for serine starvation that is essential for biofilm formation. How is intracellular serine concentration regulated during biofilm formation? Do cells down-regulate serine biosynthesis during the transition to stationary phase? Do the authors see any transcriptional down-regulation of serine biosynthesis genes? This should be clarified. In addition, the authors mention that serine is among the first amino acids exhausted from the laboratory medium during exponential growth in complex medium. How does that apply to bacteria in non-laboratory conditions, where the presumed mechanism evolved*?

Our RNA-Seq measurements did not reveal an obvious smoking gun for how serine levels might be regulated during biofilm formation. Specifically, we did not observe decreased transcription of serine biosynthesis genes. Serine also serves as a substrate for a wide range of cellular processes such as for purine biosynthesis, biosynthesis of several amino acids and phospholipid biosynthesis. However, none of these processes were transcriptionally up-regulated upon biofilm entry. In addition, conversion of serine to pyruvate is known to be catalyzed by the SdaA/B enzymes in *E. coli* upon oxygen limitation. Nevertheless, the homologous *B. subtilis* enzymes SdaAA/AB were not transcriptionally induced upon biofilm entry. Serine levels are likely regulated by an unidentified metabolic switch that accompanies biofilm entry. Future studies using metabolic labeling with stable isotope tracers could potentially shed light on this metabolic switch. For the sake of clarity, we have added a sentence to our main text that our inability to directly measure intracellular serine levels prevents us from ruling out unlikely possibilities, such as serine sequestration or changes in serine tRNA aminoacylation, which could theoretically occur in the absence of serine starvation.

We speculate on two plausible natural environments that could underlie the evolution of a codon-based serine-sensing mechanism in bacteria. In the case of enteric bacteria such as *E. coli*, alternating conditions of feast and famine could result in periodic depletion of extracellular serine, similar to that observed in laboratory complex media. In the case of free-living bacteria such as *B. subtilis*, intracellular serine levels could fluctuate as a result of alternating planktonic and biofilm lifestyles without changes in the extracellular serine levels.